## PERSPECTIVE

# In autonomic adaptation the details matter

Caitlin R. Ritchey [ID]
and James H. Peters [ID]

*Department of Integrative Physiology and Neuroscience, College of Veterinary Medicine, Washington State University, Pullman, Washington, USA*

Email: james_peters@wsu.edu

Handling Editors: Vaughan Macefield & Beth Habecker

The peer review history is available in the Supporting Information section of this article (https://doi.org/10.1113/ JP290612#support-information-section).

Neurophysiological mechanisms underlying autonomic reflexes are frustratingly elusive to pin down and often vary depending on many parameters, from time of day to feeding status to even the animal model used. In this issue Luz et al. bring some clarity to this frustration in their pursuit of delineating the brainstem adaptations to hypoxia (Luz et al., 2025). They found fundamental differences in the neurophysiological adaptations to sustained hypoxia between mouse and rat animal models. Whereas in rats suppressed oxygen results in postsynaptic changes, dependent on astrocytic mechanisms, mice also exhibit postsynaptic changes, which are independent of astrocytes. These findings seem destined to explain piles of confusing findings across researchers using rats and mice to pursue an understanding of the impact of hypoxia and the respiratory circuits. Yet they also raise the larger question of generalizability. Essentially when can we know we are pursuing a common mechanism, seen across mammals (in particular humans), and when might our mechanisms be tailored specifically to our animal model and its specific physiological demands? Any answer to this problem will need to be able to account for both the specific and general nature of physiological adaptability and the autonomic nervous system.

A key aspect of the autonomic nervous system is its role in pairing visceral organ functions to ongoing physiological demand through neural reflexes that span urgencies, magnitudes of response and timeframes. For example the baroreflex control of heart rate diligently responds within seconds, whereas gastrointestinal reflexes (with some exceptions) are satisfied within tens of seconds to minutes and even hours. A host of dedicated, but nonetheless, very similar primary afferents convey a wide array of sensory modalities to an, at best, 'loosely' organized central nucleus of the solitary tract (NTS), with all the precision expected to keep the organ systems in line (Zhao et al., 2022) – not to mention the wide range of tolerances for responding to monitored physiological parameters, from subtle beat-to-beat changes in blood pressure to meal-sized gastric distension and the associated activation of stretch receptors. In sum the autonomic nervous system is surprisingly versatile, an important quality given the myriad of physiological demands it responds to. In complement to this versatility is the ability to adapt to both acute and chronic changes in physiological parameters, such as nutrient status, blood pressure and oxygen saturation.

Thankfully neural controls that serve to monitor and keep regulated physiological end-points in range show adaptability even though these adaptations often have longer-term negative consequences secondary to the regulated parameter. This adaptability is seen in the decrease in gastrointestinal afferent sensitivity after a chronic high-fat diet (Daly et al., 2011), in the shift of baroreflex sensitivity to adjust for progressive hypertension and in the synaptic potentiation seen in intermittent and chronic hypoxia (Kline, 2010). These adaptations are well documented in rodent models (often with detailed mechanisms) and are also seen in humans (with fewer known mechanisms). Given the evidence for homology of the primary afferents, brainstem and autonomic preganglionic neurons, it is logical to conclude that mechanisms reported in one species will be similar to those in other species. And this is in large part true. We think anyway. The challenge with this conclusion is that it seems to matter at what level of analysis we are investigating. Seemingly the closer we look the better we are at seeing the differences.

This can be seen in the current findings reported by Luz and colleagues as well (Luz et al., 2025). In broad strokes the physiological adaptation to hypoxia was intact and robust between rats and mice. Chronic hypoxia potentiates synaptic transmission and NTS excitability in both rats and mice. Yet at the cellular and neurophysiological levels, important differences arise. In this regard it seems that physiological flexibility and adaptation are the selected traits, and the mechanistic 'how' the specific species achieve this outcome is less important. In an analogous fashion ion channels are well known to exhibit this inter-changeability (within reason) and are well documented to be interchangeable when needed (Vandewauw et al., 2018), often creating the same problem of determining specific *versus* general causality.

Clearly, and as always, more work is needed to be sure about this mechanism and others as they pertain to adaptability of the autonomic nervous system and physiological control. But where does this leave us who are interested in both fundamental mechanisms and generalizability? Perhaps the combination of continuing to frame our questions in the context of physiological principles, maintaining experimental control and supporting careful and accurate interpretations will be effective more often than not. The current work cautions that mechanisms are not guaranteed to be common and that when chasing an understanding of autonomic adaptation (Luz et al., 2025), the details matter.

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

## Additional information

### Competing interests

None declared.

### Author contributions

J.H.P.: conception or design of the work; drafting the work or revising it critically for important intellectual content; final approval of the version to be published; agreement to be accountable for all aspects of the work. C.R.R.: conception or design of the work; drafting the work or revising it critically for important intellectual content; final approval of the version to be published; agreement to be accountable for all aspects of the work.

### Funding

None.

### Keywords

adaptation, autonomic, hypoxia, synaptic mechanisms

### Supporting information

Additional supporting information can be found online in the Supporting Information section at the end of the HTML view of the article. Supporting information files available:

**Peer Review History**

