## [Peer Review History · The Journal of Physiology]

In autonomic adaptation the details matter.

James Henry Peters and Caitlin Rose Ritchey

DOI: 10.1113/JP290612

Corresponding author(s): James Peters (james_peters@wsu.edu)

The following individual(s) involved in review of this submission have agreed to reveal their identity: Benedito H. Machado (Referee #1)

Review Timeline:

Submission Date:	09-Dec-2025
Editorial Decision:	11-Dec-2025
Revision Received:	14-Dec-2025
Accepted:	16-Dec-2025

Senior Editor: Vaughan Macefield

Reviewing Editor: Beth Habecker

Transaction Report:

Dear Dr Peters,

Re: JP-P-2025-290612 "In autonomic adaptation the details matter" by James Henry Peters and Caitlin Rose Ritchey

Thank you for submitting your manuscript to The Journal of Physiology. It has been assessed by a Reviewing Editor and by 1 expert referee and we are pleased to tell you that it is acceptable for publication following satisfactory revision.

The review comments are copied at the end of this email.

Please address all the points raised and incorporate all requested revisions or explain in your Response to Referees why a change has not been made. We hope you will find the comments helpful and that you will be able to return your revised manuscript within 2 weeks. If you require longer than this, please contact journal staff: jp@physoc.org.

REVISION CHECKLIST:

We look forward to receiving your revised submission.

Yours sincerely,

Vaughan Macefield
Senior Editor
The Journal of Physiology

REQUIRED ITEMS

- Please ensure that the Article File you upload is a Word file.

- The reference list must be in alphabetical order, rather than numbered, to comply with our Journal format.

EDITOR COMMENTS

Reviewing Editor:

Comments to the Author:

Thank you for writing this nice perspectives piece. Please see the comments from the referee and consider making some minor edits in response.

Senior Editor:

Comments to the Author:

Thank you for submitting your interesting Perspectives article to The Journal of Physiology. It has been reviewed by the senior author of the original paper and the reviewing editor and I am pleased to report that it is potentially acceptable following satisfactory revision. I look forward to receiving your revised version shortly.

REFeree COMMENTS

Referee #1:

In the fourth line of the first paragraph of the Perspective by Peters and Ritchey they stated: They found fundamental differences in the neurophysiological adaptations to chronic hypoxia between mouse and rat animal models. Whereas in rats, suppressed oxygen results in presynaptic changes in primary afferent performance via astrocytes, mice use a different mechanism and exhibit postsynaptic changes in membrane excitability and glutamate receptor expression. In these two sentences there are some misinterpretations of our findings in rats and also in relation to our interpretation of the findings in the NTS of mice. The major points are the following: 1) in rats we documented changes in astrocytic modulation and postsynaptic currents (and not at presynaptic level as stated by the authors), while in mice we are showing that it is restricted to changes in postsynaptic currents without the involvement of astrocytic mechanisms. Therefore, the statement that "suppressed oxygen results in presynaptic changes in primary afferent" is not consistent with our previous findings in rats; 2) in relation to the statement by the authors that "mice use a different mechanism and exhibit postsynaptic changes in membrane excitability and glutamate receptor expression" it is also inconsistent with our interpretation: although the authors are right that "mice exhibit postsynaptic changes" we are not sure if post-synaptic changes in "glutamate receptor expression" contribute to the observed changes in the excitatory postsynaptic transmission. In fact, we are showing that the overall mechanisms underlying changes in the excitatory synaptic transmission in the NTS neurons of mice in response to SH are different in relation to rats. Although in rats it was documented changes in astrocytic modulation and postsynaptic currents, in mice we are showing that it is restricted to changes in postsynaptic currents. Our suggestions for adjustments in these two sentences are the following: They found fundamental differences in the neurophysiological adaptations to sustained hypoxia between mouse and rat animal models. Whereas in rats, suppressed oxygen results in postsynaptic changes, dependent of astrocytic mechanisms, mice also exhibit postsynaptic changes, which are independent of astrocytes.

END OF COMMENTS

Dear Colleagues,

Thank you for the opportunity to write a brief perspective and the constructive feedback. Below we have detailed the changes to the draft. I have also updated the reference format as per the journal standards. Please let us know if you see other edits that are needed.

Sincerely,

Jim

EDITOR COMMENTS

Reviewing Editor:

Comments to the Author:

Thank you for writing this nice perspectives piece. Please see the comments from the referee and consider making some minor edits in response.

Senior Editor:

Comments to the Author:

Thank you for submitting your interesting Perspectives article to The Journal of Physiology. It has been reviewed by the senior author of the original paper and the reviewing editor and I am pleased to report that it is potentially acceptable following satisfactory revision. I look forward to receiving your revised version shortly.

Thank you for the positive feedback. We have detailed our changes to the draft below.

REFEREE COMMENTS

Referee #1:

In the fourth line of the first paragraph of the Perspective by Peters and Ritchey they stated: They found fundamental differences in the neurophysiological adaptations to chronic hypoxia between mouse and rat animal models. Whereas in rats, suppressed oxygen results in presynaptic changes in primary afferent performance via astrocytes, mice use a different mechanism and exhibit postsynaptic changes in membrane excitability and glutamate receptor expression. In these two sentences there are some misinterpretations of our findings in rats and also in relation to our interpretation of the

findings in the NTS of mice. The major points are the following: 1) in rats we documented changes in astrocytic modulation and postsynaptic currents (and not at presynaptic level as stated by the authors), while in mice we are showing that it is restricted to changes in postsynaptic currents without the involvement of astrocytic mechanisms. Therefore, the statement that "suppressed oxygen results in presynaptic changes in primary afferent" is not consistent with our previous findings in rats; 2) in relation to the statement by the authors that "mice use a different mechanism and exhibit postsynaptic changes in membrane excitability and glutamate receptor expression" it is also inconsistent with our interpretation: although the authors are right that "mice exhibit postsynaptic changes" we are not sure if post-synaptic changes in "glutamate receptor expression" contribute to the observed changes in the excitatory postsynaptic transmission. In fact, we are showing that the overall mechanisms underlying changes in the excitatory synaptic transmission in the NTS neurons of mice in response to SH are different in relation to rats. Although in rats it was documented changes in astrocytic modulation and postsynaptic currents, in mice we are showing that it is restricted to changes in postsynaptic currents. Our suggestions for adjustments in these two sentences are the following: They found fundamental differences in the neurophysiological adaptations to sustained hypoxia between mouse and rat animal models. Whereas in rats, suppressed oxygen results in postsynaptic changes, dependent of astrocytic mechanisms, mice also exhibit postsynaptic changes, which are independent of astrocytes.

Thank you for catching these important details. I am sure I was trying to broadly state that 'other groups' have seen many mechanisms including presynaptic changes, but the wording was crude and I didn't reference appropriately to clarify. So instead, it read awkwardly. I certainly appreciate the suggested edited and have now included it in the draft.

Dear Dr Peters,

Re: JP-P-2025-290612R1 "In autonomic adaptation the details matter." by James Henry Peters and Caitlin Rose Ritchey

We are pleased to tell you that your paper has been accepted for publication in The Journal of Physiology.

Please note that Perspective articles are not typically covered by institutional open access agreements with our publisher, Wiley. Wiley do not offer article processing charge (APC) discounts for smaller article types in hybrid subscription journals, meaning that if you wish for your Perspective to be published Open Access, you will have to pay the full APC. As such, we recommend authors publish Perspectives 'behind the paywall', where they will become freely accessible after a 12-month embargo (i.e. please select the NON open access option via Wiley Author services during proofing).

Should you wish to pay for Open Access, you will be able to place an order by logging into Wiley Author services.

Yours sincerely,

Vaughan Macefield
Senior Editor
The Journal of Physiology

IMPORTANT POINTS TO NOTE FOLLOWING ACCEPTANCE OF YOUR PAPER:

- **IMPORTANT NOTICE ABOUT OPEN ACCESS:** To assist authors whose funding agencies mandate immediate public access to published research findings, The Journal of Physiology allows authors to pay an Open Access (OA) fee to have their papers made freely available immediately on publication.

- You can help your research get the attention it deserves! Check out Wiley's free Promotion Guide for best-practice recommendations for promoting your work at: www.wileyauthors.com/eoo/guide. You can learn more about Wiley Editing Services which offers professional video, design, and writing services to create shareable video abstracts, infographics, conference posters, lay summaries, and research news stories for your research at: www.wileyauthors.com/eoo/promotion.

- If you would like to receive our 'Research Roundup', a monthly newsletter highlighting the cutting-edge research published in The Physiological Society's family of journals (The Journal of Physiology, Experimental Physiology, Physiological Reports, The Journal of Nutritional Physiology and The Journal of Precision Medicine: Health and Disease), please click this link, fill in your name and email address and select 'Research Roundup': <https://www.physoc.org/journals-and-media/membernews>

EDITOR COMMENTS

Reviewing Editor:

Comments to the Author:
Thank you for making changes to your piece.

Senior Editor:

Comments to the Author:
Thank you very much for attending to the reviewer's comments. I am pleased to report that your manuscript is now considered acceptable for publication in The Journal of Physiology.

REFEREE COMMENTS

Referee #1:

The authors made the suggested adjustments in the text and the revised version of this "Perspective Article" is fine for publication in the Journal of Physiology.